# Sphingolipid-Based Synergistic Interactions to Enhance Chemosensitivity in Lung Cancer Cells

**DOI:** 10.3390/cells12222588

**Published:** 2023-11-08

**Authors:** Susana Mesén-Porras, Andrea Rojas-Céspedes, José Arturo Molina-Mora, José Vega-Baudrit, Francisco Siles, Steve Quiros, Rodrigo Mora-Rodríguez

**Affiliations:** 1Research Center on Tropical Diseases (CIET), Faculty of Microbiology, University of Costa Rica, San José 11501-2060, Costa Rica; susana.mesen@ucr.ac.cr (S.M.-P.); andrea.rojascespedes@ucr.ac.cr (A.R.-C.); jose.molinamora@ucr.ac.cr (J.A.M.-M.); steve.quiros@ucr.ac.cr (S.Q.); 2Research Center on Surgery and Cancer (CICICA), Campus Rodrigo Facio, University of Costa Rica, San José 11501-2060, Costa Rica; francisco.siles@ucr.ac.cr; 3Master Program in Microbiology, University of Costa Rica, San José 11501-2060, Costa Rica; 4National Laboratory of Nanotechnology (LANOTEC), National Center of High Technology (CeNAT), Pavas, San José 1174-1200, Costa Rica; jvegab@gmail.com; 5Pattern Recognition and Intelligent Systems Laboratory (PRIS-Lab), Department and Postgraduate Studies in Electrical Engineering, University of Costa Rica, San José 11501-2060, Costa Rica

**Keywords:** cancer, sphingolipids pathway, BODIPY, synergisms, autophagy

## Abstract

Tumor heterogeneity leads to drug resistance in cancer treatment with the crucial role of sphingolipids in cell fate and stress signaling. We analyzed sphingolipid metabolism and autophagic flux to study chemotherapeutic interactions on the A549 lung cancer model. Loaded cells with fluorescent sphingomyelin analog (BODIPY) and mCherry-EGFP-LC3B were used to track autophagic flux and assess cytotoxicity when cells are exposed to chemotherapy (epirubicin, cisplatin, and paclitaxel) together with sphingolipid pathway inhibitors and autophagy modulators. Our cell model approach employed fluorescent sphingolipid biosensors and a Gaussian Mixture Model of cell heterogeneity profiles to map the influence of chemotherapy on the sphingolipid pathway and infer potential synergistic interactions. Results showed significant synergy, especially when combining epirubicin with autophagy inducers (rapamycin and Torin), reducing cell viability. Cisplatin also synergized with a ceramidase inhibitor. However, paclitaxel often led to antagonistic effects. Our mapping model suggests that combining chemotherapies with autophagy inducers increases vesicle formation, possibly linked to ceramide accumulation, triggering cell death. However, the in silico model proposed ceramide accumulation in autophagosomes, and kinetic analysis provided evidence of sphingolipid colocalization in autophagosomes. Further research is needed to identify specific sphingolipids accumulating in autophagosomes. These findings offer insights into potential strategies for overcoming chemotherapy resistance by targeting the sphingolipid pathway.

## 1. Introduction

As one of the leading causes of mortality, cancer constitutes an important health burden worldwide [1,2] associated with aging and population lifestyle [3]. Cancer is associated with the unregulated proliferation of tumor cells; rather than responding appropriately to the signals that control normal cell behavior, cancer cells grow and divide uncontrolled [4]. Indeed, cancer cells can invade normal tissues and organs and eventually spread throughout the body [4,5]. The non-specific systemic inflammation from metastases is the major cause of mortality in cancer [1].

During this dynamic disease, cancers generally become more heterogeneous. As a result of this heterogeneity, the tumor might include a diverse collection of cells harboring distinct molecular characteristics [6,7]. Cellular heterogeneity confers tumor robustness against chemotherapy, and this property of chemoresistance continues to be the main obstacle to the effective treatment of cancer patients [8,9]. In addition, the DNA sequence can be changed due to copying errors introduced by DNA polymerases during replication. The importance of DNA damage and repair became evident because carcinogens are mutagens and cause a change in the DNA sequence [10]. Thus, carcinogens target the genetic material, contributing to the evolution of cancer cells toward more malignant phenotypes [11].

Lung cancer cells present defects in regulating normal cell proliferation and homeostasis because of a series of genetic and epigenetic alterations that trigger the processes of invasion, metastasis, and resistance to cancer therapy [12,13]. Additionally, due to its incidence and mortality, this kind of cancer has been the most common cancer worldwide since 1985 [14,15]. Therefore, we are interested in studying lung cancer cell lines as experimental models to identify potential strategies to overcome chemotherapy resistance, focusing on the sphingolipid pathway.

Indeed, sphingolipids (SLs) are one of the major classes of lipids that are present in the plasma membrane of eukaryotic cells [16]; the SL has a hydrophobic region of a sphingoid long-chain base and a fatty acid linked by an amide bond [17]. The sphingolipid signaling pathway (SL pathway) is a complex biological system that integrates different cellular stress signals and reports dynamic phenotypes related to the induction of cell death pathways [18]. Thus, the balance of certain SLs defines cell fate, which plays an important role in various aspects of cancer biology, including apoptosis, cell proliferation, cell migration, senescence, and inflammation [19,20,21]. 

The major metabolites involved in the regulation of mechanisms related to cancer biology are ceramide (Cer), sphingosine (So), and its phosphorylated metabolites ceramide 1-phosphate (C1P) and sphingosine 1-phosphate (S1P), respectively [22,23]. The balance between pro-death and pro-survival metabolites determines the cell fate, and the system that keeps this balance is called SL-rheostat [24,25]. Several studies report that Cer, So, S1P, and glucosylceramide (GluCer) are regulators in apoptosis, cell proliferation, cell migration, senescence, or inflammation processes [26]. So and Cer promote signaling mechanisms associated with cell death, such as apoptosis induction, cell cycle arrest, cell differentiation, and senescence, so that So and Cer are categorized as anti-tumor or pro-cell death sphingolipids [16,27]. Meanwhile, the glycosylation of Cer and the consequent formation of GluCer has been established as a marker of multidrug resistance [28], and it functions as a pro-survival sphingolipid [16,27]. In contrast, S1P and C1P are potent molecules inducing cell proliferation [18,22]. 

The changes can strongly influence this SL pathway’s availability of enzyme substrates in several compartments, such as the lysosome [29]. It therefore can be potentially altered by the autophagic flux of sphingolipids. Autophagy involves enclosing damaged proteins and organelles within membrane structures and delivering these molecules to the lytic compartments [30,31]. It is induced in response to starvation, reactive oxygen species, hypoxia, infection, and drugs [25,29,31]. This process is subdivided into three types: microautophagy, where lysosomes engulf cytosolic components; chaperone-mediated autophagy, which uses transport proteins for moving components into lysosome; and macroautophagy, where proteins and organelles are enclosed into autophagolysosomes (autophagosomes that merge with lysosomes), to complete the degradation [32].

Autophagy begins with the formation of the phagophore, mediated by the ULK1 complex; then, phagophore nucleation requires the phosphoinositide 3-kinase (PI3K) complex (ATG14L, VPS34, Beclin-1, and VPS15 proteins) to form the autophagosome. Autophagosomes require the lipidation of LC3 to form LC3II, and finally, the autophagosome fuses with a lysosome, forming the autophagolysosome, leading to the degradation of the cargo organelles and cytosolic proteins [25,29,31].

In addition, previous reports suggest that the progression of autophagy depends on sphingolipid production [31], attributable to its role in the formation of autophagosomes, the maturation of vacuolar proteases, or the formation of the pre-autophagosomal structure [31,33]. The role of S1P receptor expression was also linked to the regulation of autophagy; according to Ghosal et al. (2016), changes in S1P receptor expression are concurrent to the activation of autophagy, likely directing cancer cells toward death [34]. S1P receptors have been identified as mediators of autophagy and proinflammatory effects [35,36]. 

PI3K is the enzyme that produces the phosphatidylinositol 3-phosphate (PI3P) and regulates the trafficking of proteins and vesicles, and it is involved in phagophore formation. The mechanism involved in the fusion of incoming membranes to phagophore is regulated by PI3K and the Unc-51-like autophagy-activating kinase (ULK); these enzymes promote the formation and stabilization of phagophore curvature via the production of PI3P, regulating the double membrane formation [29]. Acid sphingomyelinase is mainly located within the endosomal/lysosomal compartment, and it is correlated with the cellular stress response and may become preferentially transported to the outer cell membrane [37]. Finally, the fusion of the autophagosome with the lysosome to produce the “autolysosome” is mediated by the small G protein Rab7 and cellular cytoskeleton [29].

In addition, it is known that autophagy is a metabolic process that contributes to cell death and its regulation [38]. However, it is unknown whether the regulation of cell death by autophagy is influenced by its effect on the availability of SL enzyme substrates upon changes in the flux of membranes from the autophagosomes to the lysosomes. Hence, LC3-II/LC3-PE is the only protein known to be directly associated with the autophagosomal membrane throughout the process of autophagy and thus is a widely used marker of autophagosomes [25,39]. In this study, we will analyze the behavior of autophagy modulators such as Rapamycin (RAP), Hydroxychloroquine (HCQ), Spautin (SPA), and Torin (TOR) when they are combined with chemotherapeutic drugs to treat cancer cells in the presence of a fluorescent SL biosensor to map the effects of such perturbations on the SL pathway. 

Fluorescent sphingolipids analogs (SL-analogs) have been extensively used to study the dynamics and structure of plasma membranes. BODIPY fluorophore is one of the highlight SL-analogs to study the SL synthesis and transport along secretory pathways in eukaryotic cells. Also, it is useful for visualizing membrane domains in living cells based on concentration-dependent fluorescence properties [40,41]. 

Our previous studies included in vitro and in silico assays to identify metabolic differences in the SL-rheostat of murine glioma cells with astrocytes using an SM-BOD probe (sphingomyelin labeled with BODIPY). We inferred the SL metabolic pathway topology related to the probe movement in different cell compartments [23]. Moreover, SM-BOD was used to analyze the SL distribution in plasma membranes [42]. Afterward, we constructed and fitted a dynamic mathematical model based on the metabolism kinetics of the SM-BOD probe to simulate the total fluorescent level changes upon perturbations. We combined the ordinary differential equations (ODE) to infer the metabolism of SM-BOD with Gaussian Mixture Modeling (GMM) to map how the chemotherapy is sensed in the SL pathway based on cell heterogeneity profiles [43] and a fuzzy logic model to derive rules for making predictions about drug combinations to enhance cell death and overcome thereby the drug resistance [44,45]. 

In the current report, we aimed to identify the possible mechanism of synergistic interactions of chemotherapy with inhibitors of SL-pathway enzymes and autophagy modulators imaging the behavior of a fluorescent sphingolipid analog as a biosensor to map the activity of each drug on a simple network of the SL pathway by the similarity in their perturbation-based heterogeneity profiles. Once mapped, it is possible to infer the changes in SL composition upon drug combinations to formulate a hypothesis for those synergistic interactions. Hereby, we report that epirubicin has a significant synergistic cytotoxicity when combined with the autophagy inducers Torin and rapamycin or the acid/neutral ceramidase inhibitors. The mapping of these perturbations to the SL pathway suggests that those drug combinations induce the colocalization of SLs in several cell compartments, potentially explaining the strong increase in cytotoxicity. The experimental evaluation of these interactions suggests that combining autophagy inducers with epirubicin leads to a strong colocalization of SLs. This sphingolipid-centered rationale for drug combination offers an interesting strategy to overcome drug resistance. It highlights the importance of the SL pathway as a stress biosensor for cell fate decisions.

## 2. Materials and Methods

The methodology described here is based on our previous publications (Quiros et al. (2018) and Molina et al. (2018)); however, we adapted the procedure to our cancer cell line [43,44].

### 2.1. Cell Culture for A549 Wild Type (WT)

To evaluate the SL composition, an A549 lung cancer cell line (WT) from a human lung carcinoma obtained from the NCI-60 panel was grown. The culture medium was RPMI (1640 Gibco, Life Technologies, Grand Island, NY, USA), supplemented with 1% of penicillin, streptomycin, and amphotericin B (Anti-Anti Gibco, Life Technologies, Roskilde, Denmark); L-alanil-L-glutamine 1% (GlutaMAX Gibco, Life Technologies, Grand Island, NY, USA) and fetal bovine serum 10% (FBS Gibco, Life Technologies, Eugene, OR, USA). The cultures were incubated at 37 °C (CO_2_ 5%), and trypsin (TryPLE^TM^ Express Gibco, Life Technologies, Roskilde, Denmark) was used to dissociate adherent cells.

### 2.2. mCherry-EGFP-LC3B Retroviral Vectors Assembly

Retroviral particles were generated in 60% confluent A549 cells monolayers via triple transfection with polyetilenimide (Polysciences 23966) of the transfer plasmid pBABE-puro mCherry-EGFP-LC3B [46] (provided by Jayanta Debnath, Addgene plasmid #22418), the packaging plasmid pCL-Eco (provided by Inder Verma, Addgene plasmid #12371), and the envelope plasmid pMD2.G (provided by Didier Trono, Addgene plasmid #12259) [47]. At 72 h post-transfection, virus-containing media were collected, filtered through a 0.45 µm pore membrane, supplemented with 5 µg/mL polybrene (Sigma H9268, St. Louis, MO, USA), and stored at −80 °C.

### 2.3. Generation and Sorting of Stable A549 mCherry-EGFP-LC3B Cell Line

A549 cell monolayers (80% confluency) were transduced with retroviral particles carrying the mCherry-LC3B construct and centrifuged for 2 h at 453 g at 25 °C. At 72 h post-transduction, cells were selected with 4 μg/mL puromycin (Sigma P8833) in RPMI 1640 with 10% FBS for 2 days. Cell sub-populations with homogeneous expression levels of the mCherry-LC3B construct were isolated for each cell line via fluorescence-activated cell sorting in a BD FACSJazz™ cell sorter (BD Biosciences, San Jose, CA, USA). Sorted cells were cultured in RPMI 1640 with 10% FBS with 4 μg/mL puromycin. The A549 mCherry-EGFP-LC3B cell line allowed us to track the autophagy process in the kinetic assay for the SL metabolism.

### 2.4. Evaluation of the SM-BOD Metabolism

To evaluate the kinetics, lung cancer cells (A549 WT and A549 mCherry-EGFP-LC3B) were treated with an SM-BODIPY fluorescent probe (Invitrogen™, Eugene, OR, USA), which was incorporated into the metabolism by the respective cellular machinery. In a 96-well plate, the A549 cell line was plated using a volume of 100 μL/well of cell suspension (supplemented RPMI 1640: FBS 10%, anti-anti 1%, and GlutaMAX 1%) at 1.5 × 10^5^ cells/mL. It was incubated at 37 °C for 24 h (CO_2_ 5%) to promote cellular adhesion. Then, the culture medium was removed, and 50 µL of non-supplemented RPMI (FBS 0%, anti-anti 1%, and GlutaMAX 1%) containing SM-BOD 2 µM (BODIPY FL C5 SPHINGOMYELIN Molecular Probes, Life Technologies, Eugene, OR, USA) and Hoechst 3.8 µg/mL (Molecular Probes, Life Technologies, Eugene, OR, USA) were added. Staining was incubated for 10 min at 37 °C; when the incubation time finished, 2 washes with 100 µL of non-supplemented RPMI were added immediately. 

Next, 50 µL of supplemented RPMI was added to each well of the plate with FBS 1% and propidium iodide (PI) 0.02 mg/mL. Previously, we standardized that with those concentrations of FBS and PI, the image definition obtained from the fluorescence microscopy was optimum, and the background noise was minimal. After that, we design a kinetic assay to evaluate synergisms or antagonisms of single and double perturbations. Thus, 50 μL/well of inhibitors of the sphingolipid pathway, autophagy modulators, chemotherapeutic drugs, or culture medium (controls) were added to the RPMI medium (FBS 1%, anti-anti 1%, GlutaMAX 1%, and PI 0.02 mg/mL).

Double treatments combined chemotherapies and perturbations (inhibitors of the SL pathway or autophagy modulators). The inhibitors of SL pathway enzymes were SKI (Sphingosine kinase II inhibitor, 4.13 µM, Cayman Chemical, Ann Arbor, MI, USA), PDMP (Glucosyl ceramide synthase inhibitor, 29.26 µM, ENZO Life Sciences, Farmingdale, NY, USA), DES (Desipramine, acid sphingomyelinase inhibitor and acid ceramidase, 10.32 µM, ENZO Life Sciences, Farmingdale, NY, USA), NCI (Neutral/alkaline ceramidase inhibitor, 110.62 µM, Cayman Chemical, Ann Arbor, MI, USA), D609 (Sphingomyelin synthase inhibitor, 187.62 µM, ENZO Life Sciences, Farmingdale, NY, USA), ACI (Acid ceramidase inhibitor, 5.92 µM, Cayman Chemical, Ann Arbor, MI, USA), GW (Neutral sphingomyelinase inhibitor, 1.16 µM, Cayman Chemical, Ann Arbor, MI, USA), and MYR (Myriocin, serine palmitoyltransferase inhibitor, 12.46 µM, EMD Millipore Corp, Danvers, MA, USA).

The chemotherapeutic drugs evaluated included EPI (epirubicin, 0.58 µM, Sandoz, Vienna, Austia), CIS (cisplatin, 13.32 µM, Pfizer, Sydney, Australia), and PAC (paclitaxel, 1.76 µM, AqVida, Hamburg, Germany). The autophagy modulators were RAP (Rapamycin, 1.00 µM, Sigma-Aldrich), HCQ (Hydroxychloroquine-1, 25.00 µM, Sigma-Aldrich), SPA (Spautin-2, 40.00 µM, Sigma-Aldrich), and TOR (Torin-2, 1.00 µM, Sigma-Aldrich).

To monitor the behavior of the SM-BOD sensor against different perturbations, fluorescence images were acquired using automated fluorescence microscopy (Cytation 3™, Bioteck, Winooski, VT, USA) with DAPI, Texas Red, and GFP channel every 6 h for 72 h (Figure 1). Furthermore, an automatic analysis with CellProfiler 4.1.3 software (www.cellprofiler.org 28 November 2022) allowed us to obtain 153 fluorescence features with single-cell resolution, including texture, morphology, and intensity features; moreover, this protocol estimated the cytotoxicity of each perturbation using PI stain to evaluate synergisms or antagonisms of double treatments.

According to the cytotoxicity data obtained from CellProfiler software, it was possible to obtain synergistic or antagonistic values (shown in Figure 2) using the following equations [48]:(1)$$Experimental\;cytotoxicity\left(\%\right)=\frac{dead\;cells}{total\;cells}$$
(2)$$Experimental\;viability\left(\%\right)=100-\left(Experimental\;cytotoxicity\%\right)$$
(3)$$Expected\;viability\left(\%\right)=\frac{{single\;viability}_{1}\left(\%\right)\times {single\;viability}_{2}\left(\%\right)}{100}\;\mathrm{Single}\;\mathrm{viability}\;1:\;\mathrm{Single}\;\mathrm{interaction}\;\mathrm{for}\;\mathrm{the}\;\mathrm{chemotherapeutic}\;\mathrm{drug}\;\mathrm{Single}\;\mathrm{viability}\;2:\;\mathrm{Single}\;\mathrm{interaction}\;\mathrm{for}\;\mathrm{the}\;\mathrm{inhibitor}\;\mathrm{of}\;\mathrm{the}\;\mathrm{SL}\;\mathrm{pathway}$$
(4)$$Double\;interaction\left(\%\right)=\left(Expected\;viability\right)-\left(Experimental\;viability\right)$$
(5)$$Interaction\;ratio=\frac{Double\;interaction\left(\%\right)}{100}$$

### 2.5. Model Topology for the Metabolic Fate of SM-BOD

A support vector machine algorithm in R Studio 8.13 Software was applied to reduce the features related to the fluorescence probe and nuclear attributes from the experimental protocol. The algorithm allowed the extraction and organization of the features according to the significance level. 

With the selected features, a Gaussian Mixture Model (GMM, Figure 3) was applied in the MATLAB 9.8 software [23] to identify the behavior of chemotherapeutic drugs on the SL pathway, comparing their effect on the cellular heterogeneity profile with the effect produced by known inhibitors of the SL metabolic pathway [43].

The cellular heterogeneity profile provided the information to perform clusters and to associate different perturbations in a dendrogram, in which it was possible to define double interactions to identify synergisms and antagonisms. That profile allows us to infer how the chemotherapeutic agents and inhibitors perturb the SL pathway for proposing a network topology of the SL pathway for the A549 lung cancer cell line based on the metabolism of the SM-BOD probe (Figure 4).

## 3. Results

### 3.1. Chemotherapy Interacts with Autophagy Modulators and Inhibitors of the Sphingolipid Pathway to Modify Cytotoxicity

To develop the kinetic assays for feature extraction with single-cell resolution, we designed a protocol of image segmentation to evaluate the effect of single and double perturbations of the SL pathway (inhibitors or autophagy modulators and chemotherapeutic drugs) in single cells, using the CellProfiler software. Figure 1 shows the segmented images using fluorescence microscopy for the lung cancer cell model; cell nuclei recognized with Hoechst and propidium iodide (PI) probes (Figure 1A) and the presence of fluorescent sphingomyelin probe that was analyzed in the GFP channel (shown in Figure 1B), both enabled us to analyze cell viability as it is shown in Figure 1C. These results suggest perturbation-associated changes in the SL metabolism that can be monitored with single-cell resolution, using a PI probe to report cell death and a sphingomyelin fluorescent analog to track SL metabolism.

Then, we calculated cell viability from the image data of the single and double interactions of the previous kinetic assays (Section 2 and Section 2.4, Equations (1)–(5)). The expected viability and experimental viability data are shown in Figure 2A,B, respectively, to identify synergistic and antagonistic interactions for double treatments of chemotherapy-inhibitor perturbations (Figure 2C).

The results associated with EPI indicate that the chemotherapy-modulator interaction of EPI with RAP showed an important synergism leading to a decrease in viability to 14% (Figure 2B) compared to the expected combined viability without interaction of around 67% (Figure 2A). According to these data, we obtained a synergistic effect of 0.5-fold, the higher interaction reported in our results. Additionally, the synergism of EPI-TOR and EPI-ACI was 0.4 for both cases, and the interaction of EPI-NCI was 0.3. In this way, we suggest that the mechanism of action of EPI interacts at the pathway level with the mTOR inhibitor protein (TOR-RAP) and with the ceramidase inhibitor (ACI-NCI), leading to a significant synergistic effect. Several weaker synergistic interactions were reported for EPI with SKI, DES, D609, GW, MYR, HCQ, and SPA. The double treatment of EPI-PDMP reported no interaction, and no antagonism was observed in our results for epirubicin.

In addition, the chemotherapeutic drug cisplatin (CIS) showed the main synergism with the neutral ceramidase inhibitor (NCI) of a 0.3-fold increase compared to the expected cytotoxicity; the other conditions showed weaker synergistic interactions. A no-interaction result was obtained for CIS-D609, and no antagonism was reported in our results.

The analysis with paclitaxel (PAC) demonstrated that most interactions showed antagonistic effects, with the strongest antagonistic interaction reported to be −0.4 with SPA. Two weak synergisms with DES and GW (0.2 and 0.1, respectively) were also detected.

### 3.2. A GMM Model of Heterogeneity Profiles of the Metabolic Fate of a Fluorescent Sphingomyelin Analog Suggests the Mechanistic Base of the Synergistic Interactions Mapped to the SL Pathway

A support vector machine algorithm (R Software) was applied to select and organize the features extracted from the imaging analysis (Section 2 and Section 2.5). A Gaussian Mixture Model (GMM, MATLAB Software) was performed to establish a heterogeneity profile to infer the effect of unknown perturbations on the SL pathway (chemotherapy). We used 109 selected fluorescence features out of 153 total features extracted by imaging. The cellular datasets for all conditions were merged into a single dataset to identify the maximum number of normally distributed subpopulations [43] (Figure 3A–D). For this purpose, we identified an optimal number of nine significant subpopulations by MANOVA test (*p* < 0.5, Figure 3A,B). Thereby, a heterogeneity profile showed the frequency of cells (Figure 3F) clustering to each of those subpopulations with the profile produced by a known perturbation (Figure 3E). EPI-ACI, EPI-NCI, CIS-PDMP, HQC-SPA, and RAP-TOR were the chemotherapy-inhibitor interactions that could be mapped together due to the similarities in their heterogeneity profiles. This enabled us to suggest that the influence of those chemotherapeutic agents on the metabolic pathway of SM-BOD is like those corresponding inhibitors of the SL-pathway enzymes.

Next, we propose a network topology of the SL pathway for the A549 lung cancer cell line (Figure 4) based on the metabolism of the SM-BOD probe that we described in previous studies [44,45]; the green boxes represent the fluorescent metabolites, the white ones indicate loss of fluorescence and magenta boxes are the enzymes-inhibitors of the SL pathway. We coupled this model topology of the autophagic process with its main steps enclosed in dotted circles (P: phagophore, AP: autophagosome, and AL: autophagolysosome).

According to the results obtained from the GMM, we associated the chemotherapy-inhibitor pairs (obtained from Section 3 and Section 3.2) to map their interactions to the SL pathway. The yellow boxes represent the chemotherapeutic drugs sensed according to the clusters identified on the dendrogram plot (Figure 3E), which were EPI-NCI, EPI-ACI, and CIS-PDMP. RAP-TOR and SPA-HCQ autophagy modulators (blue and grey boxes, respectively; Figure 4) were associated. Still, it was impossible to map them to a known perturbation within the SL pathway. Taken together, those results suggest that the chemotherapeutic treatments perturb the SL pathway similarly to their corresponding known inhibitors. 

Furthermore, our model topology suggests potential mechanisms of those double perturbations identified to have synergistic effects in cytotoxicity. The EPI-ACI and EPI-NCI pairs potentially inhibit the synthesis of sphingosine (So) and enhance the concentration of ceramide (Cer) in different cellular compartments. The inhibition of neutral ceramidase by the EPI-NCI pair potentially increases the Cer content locally at the plasma membrane (CER_PM). Meanwhile, the inhibition of acid ceramidase enzyme by EPI-ACI interaction leads to Cer accumulation in the lysosomal compartment. Moreover, the interaction of CIS-PDMP potentially inhibits the glucosylceramide synthase to avoid the synthesis of glucosylceramide (GLU_CER), leading to an increase in Cer in the endoplasmic reticulum and on the cytosolic side of the Golgi compartment.

Our model topology also suggests that the interactions of chemotherapy with TOR or RAP autophagy inducers stimulate the formation of autophagosomes to move cellular material (including sphingolipids) from the plasma membrane (SM_PM) to the lysosomal compartment (CER_LYS). 

Like the aforementioned, the cluster of the autophagy inhibitors SPA-HCQ did not map to any known perturbation of the SL pathway. According to our network model, SPA inhibits the formation of the phagophore in the plasma membrane, and interestingly, HCQ shows similar behavior to ACI, thereby inhibiting the acid ceramidase enzyme, potentially preventing the formation of autophagolysosomes. 

These results indicate that the heterogeneity profiles obtained from a GMM model of the image features derived from the metabolic fate of a fluorescent sphingomyelin analog enable the mapping of the influences of single perturbations on the SL pathway. This topological mapping also enables us to suggest the mechanistic base of the synergistic interactions between chemotherapeutic drugs and known perturbations of autophagy and SL pathway.

### 3.3. mCherry-EGFP-LC3B and SM-BOD Cell Labeling Show That the Interactions of Epirubicin with Torin and Rapamycin Stimulate the Accumulation of Autophagosomes

According to our model topology, we hypothesized that interactions of chemotherapeutic drugs with the autophagy inducers TOR and RAP lead to an increase in vesicles potentially associated with the local accumulation of ceramide in those compartments (Figure 4). In order to confirm that hypothesis, we designed a composite image of control, double, and single perturbations for the interactions of EPI-RAP and EPI-TOR in stable A549 mCherry-EGFP-LC3B cells labeled with SM-BOD (Figure 5) in order to visualize the presence of fluorescently labeled (SL-loaded) autophagosomes for our experimental conditions, using fluorescence microscopy.

The composite image of the A549 mCherry-EGFP-LC3B cells (Figure 5) indeed shows a significant increase in vesicles (the dotted pattern) for both synergistic conditions of EPI-RAP and EPI-TOR compared to their corresponding single perturbations and control at 24 h. These results unveil a potential mechanism of synergism by the combination of autophagy inducers with inhibitors of lysosomal sphingolipid metabolism (or chemotherapeutic drugs with a similar effect), leading to a local increase in sphingolipids in those autophagosomal compartments inducing cell death.

## 4. Discussion

Sphingolipids are essential structural components of biological membranes and are crucial regulators of cellular processes [49]. Some SLs behave as second messengers involved in cell fate decisions, such as Cer (involved in cell death) and S1P (promotes cell survival) [50]. In the present work, a fluorescent sphingomyelin analog probe enabled cell segmentation by imaging and the determination of the level of chemosensitivity in the A549 lung cancer cell line treated with different perturbations of the SL pathway. The cell viability analysis defined the synergistic interactions that enhance or reduce the cytotoxic effect on this cell line. Furthermore, we proposed a model topology with mapped perturbations of the identified synergies to propose potential mechanisms explaining the metabolic fate of the fluorescent biosensor. In addition, the composite images of the kinetic assays evidence the perturbation to the autophagic process for some specific conditions where an increase in autophagosomes is observed with a concomitant colocalization of fluorescent SL.

The BODIPY is characterized as being one of the most useful fluorophores due to its chemical and thermal stability and its properties as a fluorescent sensor to interact directly with different proteins and peptides [51]. We suggest that the use of BODIPY to label a sphingolipid analog and track their cell metabolism is a promising strategy to find synergistic interactions between chemotherapeutic agents and autophagy modulators or inhibitors of the SL pathway to enhance or reduce cytotoxicity.

The SM-BOD sensor is located mainly in the plasma membrane. As time goes on, it is internalized toward the Golgi apparatus, increasing the signal at the perinuclear zone and the affinity of Cer to this organelle (mediated by the interconversion from SM to Cer) [52], which confirms the potential of SM-BOD to track the sphingolipid metabolism in cells (Figure 1). According to the synergies found in our results (Figure 2 and Figure 3), we proposed a cellular model for the A549 lung cancer cell line to analyze the SM-BOD metabolism, including autophagic compartments, to dissect the interaction between those two important cellular processes in the context of double perturbations including chemotherapy and autophagy modulators (Figure 4).

The anthracycline EPI is a DNA-intercalating genotoxic drug used for cancer treatments [53], including lung cancer. This chemotherapeutic drug has many side effects related to its high toxicity and the emergence of chemoresistant neoplastic clones. Interestingly, anthracycline resistance is associated with an adjustment in the metabolism of lipids and the autophagic process [54,55]. In addition, EPI treatment modulates the SL metabolic pathway and regulates autophagy in a cell type-specific manner [53,56]; previous reports described that anthracyclines trigger the cell death process by stimulating the production of Cer [57]. 

After the evaluation of the postulated interactions among different perturbations, we identified that EPI presents an important synergism with TOR and RAP, which are autophagy inducers. Therefore, our mathematical model indicates that those perturbations inhibit the formation of So but increase the Cer content (as shown in Figure 4). Based on scientific reports, both RAP and TOR stimulate autophagy by inhibiting the mTOR protein complex, thereby increasing the transport of cellular materials from the plasma membrane to the lysosome, including sphingolipids such as the SM-BOD [31]. That information agrees with our results because of the increase in the autophagosomal compartment (vesicles) for those conditions, indicating that the autophagic process is active and suggesting colocalization of sphingolipids (potentially Cer) in the autophagosomes due to EPI-TOR and EPI-RAP synergisms in the A549 cell line (Figure 5). 

Furthermore, we reported that EPI has synergistic interactions with the acid (ACI) and neutral (NCI) ceramidase inhibitors (Figure 2). In fact, the dendrogram shows that EPI has a similar behavior as ACI and NCI (Figure 3E). However, the perturbations of ACI could also lead to pathological changes in the absence of Cer accumulation and lead to changes in gene expression within the sphingolipid pathway [58].

Autophagic formation is a very complex process with new adaptor molecules discovered to mediate cell death and tumor metastasis [59]. Autophagy is a deep and complex process which describes initiation, nucleation, elongation, maturation, fusion, and degradation [60]. Autophagosomes are fused with lysosomes to form autophagolysosomes, which degrade their content (organelles and proteins) in order to keep the cell homeostasis and organelle renewal. The regulatory mechanisms of autophagy are complex, and its upstream signaling pathway mainly involves an mTOR-dependent pathway and mTOR-independent pathway (AMPK, PI3K, Ras-MAPK, p53, PTEN, and endoplasmic reticulum stress) [61]. 

Moreover, studies have demonstrated that the sphingolipids modulate autophagy. Reports indicate that the exposure of cancer cell lines to chemotherapeutic treatments increases Cer content [50]. Cer induces autophagy via three mechanisms: (i) the direct anchoring of autophagosomes to mitochondria [62], (ii) the down-regulation of nutrient transporters [63], and (iii) the disruption of the Beclin-Bcl-2 complex (an autophagy inhibitor) [64,65]. During autophagy, cytoplasmic constituents (protein and organelles) are engulfed in a double-membraned phagophore; once the double membrane fuses and the autophagosome matures, it is transported along tubulin tracks to a perinuclear location, fusing with a lysosome which generates an autolysosomal structure capable of degrading its content [29]. The above agrees with the mapping site of the autophagic process on the model topology shown in Figure 4, in which Cer accompanies the accumulation of autophagosomes. Furthermore, Figure 5 shows the colocalization of sphingolipids (presumably Cer) at the lysosome level (vesicles), suggesting that the enhancement in autophagosome formation by TOR and RAP and the interruption of autophagosome degradation by EPI leads to a local accumulation of sphingolipids potentially triggering cell death.

Withstanding the fact that our in silico model (shown in Figure 4) describes that the interaction of EPI-TOR and EPI-RAP promotes an accumulation of Cer in autophagosomes, our results of the kinetic analysis (Figure 5) show an important colocalization of SL (potentially Cer) in autophagosomes. Further studies are required to determine the identity of the sphingolipids accumulating within those growing autophagosomes under conditions where the autophagosomal and SL degradation are blocked.

As a potent bioactive sphingolipid implicated in the regulation of cancer progression and cellular proliferation, S1P receptors are involved in controlling tumor response to chemotherapy. S1P receptors are crucial regulators of cancer cell survival and proliferation via activation of key pathway clusters such as Ras/Erk and PI3 K/Akt [34,66]. Due to its role in the activation of autophagy, S1P and the expression of its receptors are also involved in the transactivation of growth factor signaling mechanisms, which could be a potentially interesting direction for further studies as a novel treatment approach. 

On the other hand, the chemotherapy CIS is implied in death cells in pathways regulated by sphingolipids, in which CIS increases the ceramide content [67]. In agreement with that information, our data show that the CIS drug has the main synergism with NCI (Figure 2), and in our model, it was mapped with PDMP (a glucosylceramide synthase inhibitor, Figure 4). This interaction would increase the Cer content in the cytosolic region of the Golgi apparatus (CER_GOLGIOUT), avoiding glycosylation and its conversion to So at the plasma membrane [43]. Based on Dupre et al. (2017), the administration of PDMP to lung cancer cells stimulates endoplasmic reticulum stress (ER stress), autophagy, and apoptosis as a result of PDMP-induced ceramide accumulations [67]. 

Besides the aforementioned, studies have demonstrated that PAC increases Cer content and stimulates the expression of sphingomyelinases, so that PAC promotes the hydrolysis of sphingomyelin [68]. In our results, PAC chemotherapy was able to perturb the SL pathway. Still, it could not be mapped to any known inhibitor, meaning it probably perturbs the SL pathway differently than the inhibitors used in this study. It is important to consider that the model simplifies a more complex pathway with a limited number of inhibitors. 

Previous reports indicate that SPA inhibits autophagy and avoids phagophore formation [69]. Meanwhile, HCQ inhibits autophagy due to lysosomal acidification and blocks the fusion of autophagosomes with lysosomes, triggering the accumulation of autophagosomes [70]. In contrast, our experimental data show that HCQ and SPA were mapped together in the same cluster, suggesting that they acted similarly for this cellular model. A possible explanation for this discrepancy is that both perturbations have similar consequences at the signal transduction level, and it was therefore impossible for the algorithm to separate and map them correctly. In consequence, our model shows that SPA was mapped inhibiting the phagophore, and HCQ was mapped together with ACI because the inhibition of the autophagolysosome acidification would not allow it to reach the optimal pH for the correct activation of acid ceramidases; therefore, it would have a similar role compared to this inhibitor [71] (Figure 4).

## 5. Conclusions

Chemotherapy is commonly used to treat cancer, but it has limited survival benefits due to its high toxicity [72] and the pharmacological resistance due to neoplastic development. Our study was based on the characterization of tumor heterogeneity as a tool to map the critical importance of sphingolipids in cell fate and stress signaling. Based on the results obtained from this work, we suggest a strategy for combinatorial drug treatment aimed at increasing the levels of anti-tumoral sphingolipids in critical compartments, leading to synergistic interactions to enhance the anti-neoplastic effects of chemotherapy. 

The use of fluorescent sphingolipid analogs and molecular sensors of autophagy provided valuable insights into the cytotoxic interactions of various chemotherapeutic agents, including epirubicin, cisplatin, and paclitaxel, in combination with inhibitors of the sphingolipid pathway and autophagy modulators. Our results revealed significant synergy, particularly between epirubicin and autophagy inducers (rapamycin and Torin), leading to a strong reduction in cell viability. Cisplatin also demonstrated synergistic effects when combined with a ceramidase inhibitor. However, paclitaxel often exhibited antagonistic effects when used in combination with these modifiers.

The model we proposed provided critical insights into the influence of chemotherapy on the sphingolipid pathway, suggesting that combining chemotherapies with autophagy inducers increases vesicle formation, potentially linked to ceramide accumulation, which triggers cell death.

Via an analysis of sphingolipid metabolism and autophagic flux in the context of chemotherapeutic interactions using an A549 lung cancer model, our results establish the basis for the design of new therapeutic protocols with potential clinical applications with the aim to enhance chemosensitivity in heterogeneous tumors and thus improve the efficiency and efficacy of cancer treatments.

## Figures and Tables

**Figure 1 cells-12-02588-f001:**
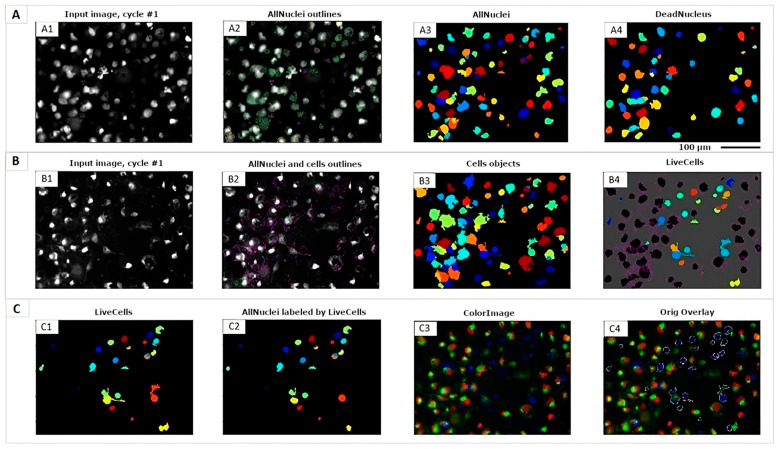
Imaging analysis for the subcellular segmentation (nuclei/cytoplasm) for the lung cancer cell model (A549 WT) by fluorescence microscopy. (**A**) Nuclei analysis stained with Hoechst and propidium iodide (PI) using DAPI and Texas Red channels, respectively; (**A1**) nuclei of all cells obtained by adding the signal emitted by DAPI and Texas Red channels; (**A2**) nuclear recognition by the protocol designed using CellProfiler software; (**A3**) nuclei identification of all detected cells; (**A4**) nuclei identification of dead cells by Texas Red channel. (**B**) SM-BOD probe analysis by GFP channel; (**B1**) sphingolipids labeled with the fluorescent probe; (**B2**) complete cells recognition by the protocol designed using CellProfiler software; (**B3**) identification of all complete cells; (**B4**) dead cell (**A4**) removal from the complete cells identified in b3 to obtain only live cells. (**C**) Cell viability analysis; (**C1**) complete viable cells; (**C2**) nuclei viable cells; (**C3**) tricolor image of signals from DAPI, GFP, and Texas Red channels for all cells; (**C4**) live cells removal from the complete cells to identify the overlay for dead cells. Scale: 100 µm.

**Figure 2 cells-12-02588-f002:**
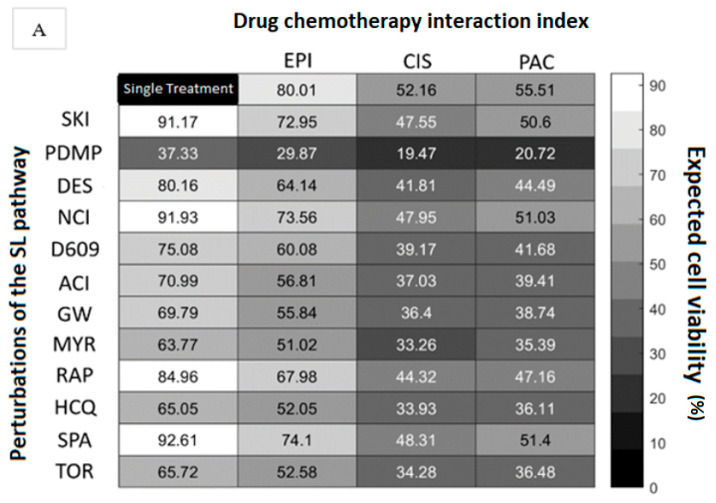
Cell viability analysis at 48 h from the A549 cell line (WT) enables the recognition of chemotherapy-inhibitor interactions. (**A**) Percentage of expected cell viability for single and double treatments; (**B**) percentage of experimental cell viability for single and double treatments; (**C**) synergistic and antagonistic interactions of chemotherapeutic drugs and perturbations of the SL pathway. EPI: Epirubicin; CIS: Cisplatin; PAC: Paclitaxel; SKI: Sphingosine kinase II inhibitor; PDMP: Glucosyl ceramide synthase inhibitor; DES: Desipramine, acid sphingomyelinase inhibitor and acid ceramidase; NCI: Neutral/alkaline ceramidase inhibitor; D609: Sphingomyelin synthase inhibitor; ACI: Acid ceramidase inhibitor; GW: Neutral sphingomyelinase inhibitor; MYR: Myriocin, serine palmitoyltransferase inhibitor; RAP: Rapamycin; HCQ: Hydroxychloroquine-1; SPA: Spautin-2; TOR: Torin-2.

**Figure 3 cells-12-02588-f003:**
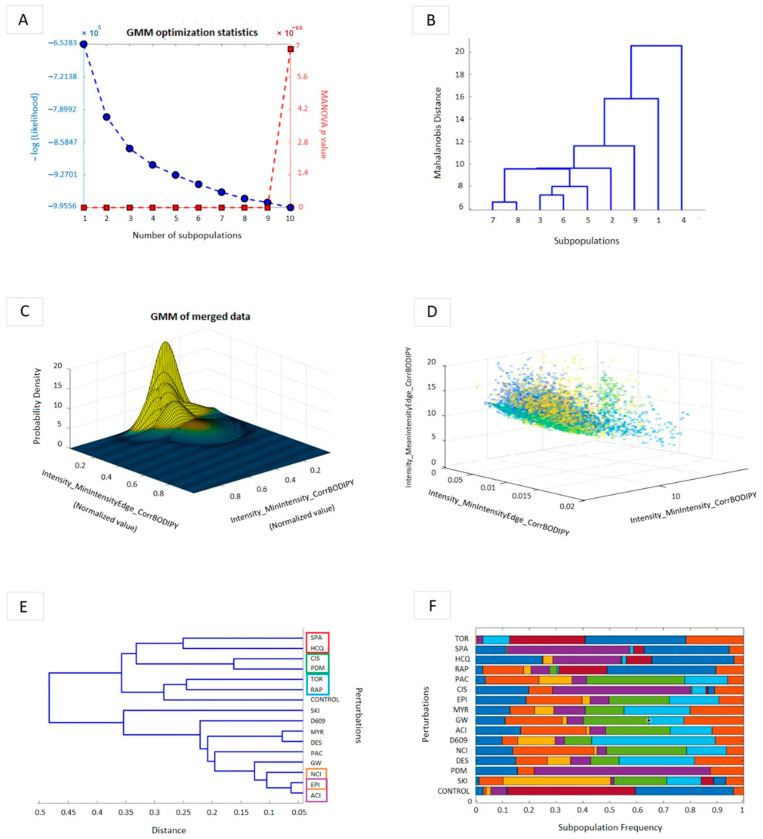
The Gaussian Mixture Model (GMM) enables the association of perturbations, potentially explaining synergistic interactions. (**A**) GMM statistical optimization using MANOVA (*p*-value) and −Log (likelihood) parameters. (**B**) Selected-subpopulations interactions. (**C**) Gaussian Mixture Model plot of merged data. (**D**) Cell sub-population classification in a 3D plot. (**E**) Dendrogram plot. The enclosed pairs are the interactions of interest. (**F**) Subpopulation frequency related to the perturbations applied. The different colors presented correspond to different cell subpopulations.

**Figure 4 cells-12-02588-f004:**
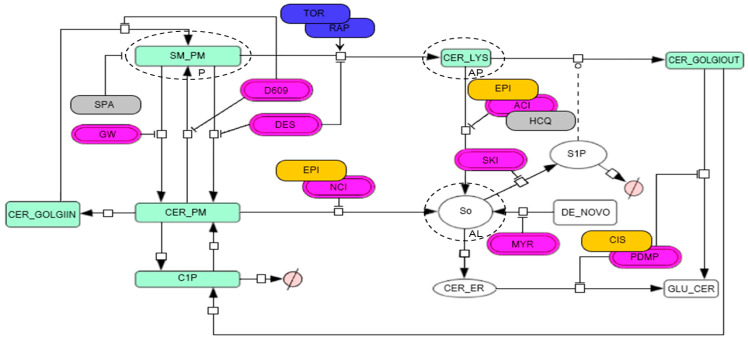
Model topology for the A549 lung cancer cell line of the metabolic fate of a fluorescent-sphingomyelin analog (SM-BOD) with the autophagic process (dotted circles; P: phagophore, AP: autophagosome and AL: autophagolysosome) allows inferring how the different chemotherapeutic agents and inhibitors perturb the SL pathway.

**Figure 5 cells-12-02588-f005:**
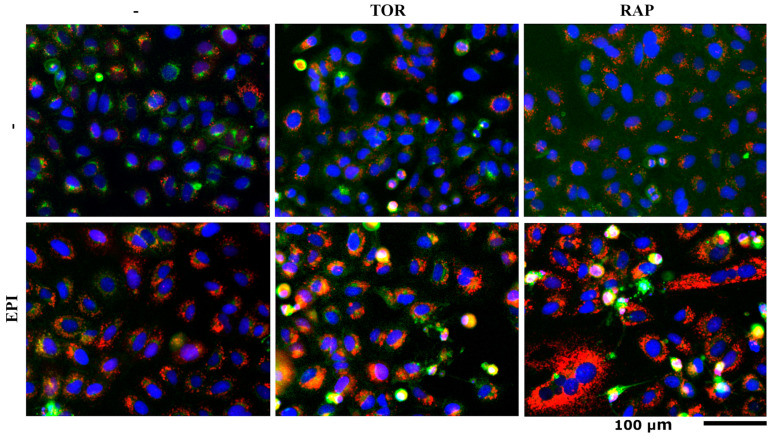
Kinetic analysis evidences the autophagy by increasing the autophagosome content for the selected synergistic conditions at 24 h for the A549 mCherry-EGFP-LC3B cells using fluorescence microscopy. Imaging composite of control, single perturbations of EPI, RAP, and TOR, and double perturbations of EPI-RAP and EPI-TOR. Scale: 100 µm.

## Data Availability

Models are available upon request.

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
