# Peer review of "Sphingolipid-Based Synergistic Interactions to Enhance Chemosensitivity in Lung Cancer Cells"

_cells, 2023, doi:10.3390/cells12222588_

Round 1
Reviewer 1 Report
Comments and Suggestions for Authors
The original research study investigated the chemotherapeutic interactions using a lung cancer model with A549 cell line. The authors characterised cytotoxic interactions between conventional chemotherapeutic drugs (epirubicin, cisplatin, and paclitaxel) and described the corresponding perturbations of the sphingolipid pathway. The study is interesting, a lot of data was generated, although there are some serious problems to address.
Major issues to address
1. Abstract does not describe the major findings, but mostly represents some methods and aims. Authors should talk about their actual data. They wrote’ we implemented a cell model approach.. etc.” but what is the outcome of this implementation? It looks unfinished and unclear. Re-writing is required.
2. Sphingolipid interactions are indicated in the title, but the Abstract does not really make a clear statement about the sphingolipid interactions: which sphingolipids? what kind of interactions were observed?
3. Introduction, line 67: citing of only one review manuscript form 2012 ( 11 year old reference) is not sufficient. I suggest citing more recent sphingolipid-related cancer-targeting reviews such as here https://pubmed.ncbi.nlm.nih.gov/31863815/
4. The Introduction has got a lot of information about autophagy, while the Abstract has none.
5. Authors wrote (line 97) “ … progression of autophagy depends on sphingolipid production [22]” it is incomplete information. The role of S1P receptor expression was also linked to the regulation of autophagy ( see here https://pubmed.ncbi.nlm.nih.gov/27261597/).
6. Line 135 “ with drugs and perturbations (inhibitors and autophagy modulators)…” sounds wrong. Should it be this way “with drugs (inhibitors (which ones?) and autophagy modulators) and perturbations (in the metabolism of SL? which ones?)…”? the sentence is too long and should be broken into at least two shorter sentences.
7. Figure 1: quantitative analysis is missing for this data. Does figure 2 represent the quantitative analysis of data shown on Fig1? If it is true, these sections should be combined and the Results subsection should be merged.
8. Section 3.3 of Results provides information that was not indicated in the Abstract clearly. Authors should make a solid statement that the novel method/model (mapping) has been developed. The outcome of this method allowed to link autophagy to SL pathway. It should be indicated.
9. Drug-induced accumulation of SL in the autophagosomes was not presented in the Abstract. It should be accented.
10. Discussion: role of S1P receptors was not discussed at all. The complexity of autophagosome formation and fusion should be at least mentioned; relevant papers should be cited ( see here https://pubmed.ncbi.nlm.nih.gov/36342046/ and https://pubmed.ncbi.nlm.nih.gov/32111095/). Potential signaling links can be indicated/stressed as targets for future studies.
11. Conclusions are too general and does not reflect the true findings properly.
Comments on the Quality of English LanguageQuality of English is Ok, but editing is required.
Reviewer 2 Report
Comments and Suggestions for Authors
Using fluorescence microscopy the authors analyze the interactions between some common drugs used for fighting cancer and sphingolipid based molecules. The signaling pathways in which sphingolipid molecules are essential actors and become altered during treatment are unveiled. The authors conclude that some drug combinations of chemotherapy and sphingolipid pathway inhibitors could overcome chemotherapy resistance.
The experimentation is well conducted according to authors' description. Both methodology and results are conveniently described and conclusions fit with the experimental results.
A few issues are suggested to be addressed before the manuscript is accepted for publication:
1) Line 150: which protocol?
2) Line 174: use g values where appropriate
3) Lines 173-180: please indicate what the difference is between mCherry-LC3B and LC3/m-Cherry A549 cell lines.
4) Equation 3: please give the definition for single viability 1 and 2
5) Why no capital letters are used in the figure legends and throughout the text as they are in the figure panels?
6) Line 368: "... an increase 2 of Cer ..." ?
7) Line 410: sphingolipids are not the major constituents of biological membranes.
Round 2
Reviewer 1 Report
Comments and Suggestions for Authors
Authors addressed nearly all my comments properly. The revised version has been improved.